# Knowledge of Community Pharmacists in Saudi Arabia Regarding Human Monkeypox, Its Management, Prevention, and Vaccination: Findings and Implications

**DOI:** 10.3390/vaccines11040878

**Published:** 2023-04-21

**Authors:** Alian A. Alrasheedy, Abdulrahman H. Aldawsari, Munyib I. Alqasir, Omar A. Alsawyan, Osama A. Alalwan, Saleh A. Alwaker, Masaad S. Almutairi, Brian Godman

**Affiliations:** 1Department of Pharmacy Practice, College of Pharmacy, Qassim University, Qassim 51452, Saudi Arabia; 371109867@qu.edu.sa (A.H.A.); 371108846@qu.edu.sa (M.I.A.); 371111565@qu.edu.sa (O.A.A.); 371109082@qu.edu.sa (O.A.A.); 361108991@qu.edu.sa (S.A.A.); mas.almutairi@qu.edu.sa (M.S.A.); 2Department of Pharmacoepidemiology, Strathclyde Institute of Pharmacy and Biomedical Sciences, University of Strathclyde, Glasgow G4 0RE, UK; brian.godman@strath.ac.uk; 3Department of Public Health Pharmacy and Management, School of Pharmacy, Sefako Makgatho Health Sciences University, Molotlegi Street, Garankuwa, Pretoria 0208, South Africa; 4Centre of Medical and Bio-Allied Health Sciences Research, Ajman University, Ajman P.O. Box 346, United Arab Emirates

**Keywords:** community pharmacists, monkeypox, mpox, non-endemic, outbreak, pandemic, Saudi Arabia

## Abstract

Many cases of monkeypox have recently been reported in countries where this disease is not endemic, raising a global health concern. Consequently, healthcare professionals (HCPs), including pharmacists, need to be aware of the disease, its prevention, including the role of vaccines, and its management to reduce transmission. A cross-sectional, questionnaire-based study was conducted among conveniently sampled community pharmacists in the Qassim region of Saudi Arabia. A total of 189 community pharmacists participated in the study, giving a response rate of 72.97%. From these, 86.77% were male, 51.32% were ≤30 years old, 36.51% were aged between 31–40 years, and 43.39% had 1–5 years of experience as community pharmacists. Their overall knowledge was 17.72 ± 5.56 out of a maximum of 28. The overall rate of correct answers for the knowledge statements was 63.29%, with 52.4% answering ≥50–<75% of the knowledge questions correctly and 31.2% answering ≥75% of the questions correctly. The knowledge subdomain related to diagnosis and clinical characteristics recorded the highest score, with the subdomain relating to causative pathogens and epidemiology recording a lower score. Overall, community pharmacists had moderate knowledge of monkeypox and its clinical management, prevention, and the role of vaccines, which is a concern for the future. Consequently, tailored, flexible, and timely educational interventions are needed to ensure that HCPs, including community pharmacists, are fully equipped with the latest evidence-based knowledge regarding this viral disease to reduce transmission and improve care.

## 1. Introduction

Monkeypox was first identified during an outbreak in an animal facility in Copenhagen, Denmark, in 1958 [1,2,3]. This initial outbreak occurred in cynomolgus (*Macaca fascicularis*) monkeys received by the facility from Singapore, which were being used for research into potential vaccines for polio. Following this, after more than ten years, the first case of monkeypox in humans was reported in 1970 in the Democratic Republic of the Congo (DRC) [1,4,5]. Human monkeypox (MPX) was not reported outside of Africa until 2003, when an outbreak of human MPX was reported in the USA. Since then, other cases and outbreaks have been reported in other countries [2,3,6,7,8]. 

The monkeypox virus (MPXV) belongs to the family *Poxviridae*, the subfamily *Chordopoxvirinae*, and the genus *Orthopoxvirus*, with two known groups of MPXV (i.e., distinct genetic clades), namely the West African and Congo Basin (Central African) clades, with the latter more virulent with an appreciable higher case fatality rate [2,4,8,9]. In August 2022, the World Health Organization (WHO) published new names for these variants/clades to be Clade I for the Central African clade and Clade II for the West African clade. Furthermore, Clade II was divided into two subclades, i.e., Clade IIa and Clade IIb [10,11,12]. The infections in the current global outbreak are caused by Clade IIb, especially in non-endemic countries [13,14].

On 23 July 2022, the WHO declared human MPX as a Public Health Emergency of International Concern (PHEIC) [15]. This was due to several outbreaks in many countries [16,17,18]. By 7 March 2023, there had been 86,396 confirmed cases reported from 110 countries with 108 deaths [19]. This is increasingly seen as a public health concern, especially as a number of cases have been reported in non-endemic countries, accounting for 102 of the reporting countries, and considering that the disease is not directly related to animal or human traveling [13,17,20]. This is in addition to the impact on morbidity and mortality in endemic countries, including the Democratic Republic of Congo [4]. In Saudi Arabia, the first case of MPX was reported on 14 July 2022 in Riyadh in an individual returning from abroad, i.e., an imported case [21], and as of 7 March 2023 there have been eight cases reported in the country [19].

The human MPX incubation period varies between 4 and 21 days [3,5,22,23,24], and during this period patients will be asymptomatic and not contagious [22,25]. The infection can subsequently be divided into two main phases, which include the invasion phase, i.e., the prodrome period, and the skin eruption phase [3,26,27]. The clinical presentation, i.e., the signs and symptoms of infected individuals during the prodromal phase, can include fever, headache, chills, malaise, myalgias, lymphadenopathy, and pain, as well as respiratory symptoms including a cough, sore throat, or nasal congestion. Skin rashes and lesions are the key characteristic of the skin eruption phase of MPX [22,25,26,27,28,29,30,31]. The skin lesions are typically more localized on the face in up to 95% of cases and palms and soles in up to 75% of cases, as well as in oral mucous membranes [3]. The lesions evolve sequentially through several stages. These range from macules and papules to vesicles and pustules, and subsequently to scabs. Once all the lesions are scabbed over and a new layer of skin is formed, the person is no longer contagious, with the illness typically lasting 2–4 weeks [3,22,23,27,29,30,31]. However, it should be noted that in the 2022 MPX outbreaks, atypical presentations were seen in which prodromal symptoms did occur after the rash or were even not present at all. In addition, the rashes seen could be limited or localized or be only a single lesion [25,28,29]. 

Many routes of transmission have been reported in the literature for this virus [4,17,32]. Animal-to-human transmission may occur via direct contact with infected animals, their body fluids, or materials, as well as through eating raw or minimally processed meats, i.e., bushmeat of infected animals. Animal bites and scratches have also been reported as potential sources of infection [32,33,34,35,36,37]. Human-to-human transmission includes direct or close contact with skin lesions, respiratory secretions, or fluids of infected persons or contaminated objects, fomites, or surfaces infected with the MPXV, in addition to via respiratory droplets, especially prolonged face-to-face contact [32,33,34,35]. In view of this, living in the same household, sharing objects or materials, or eating from the same dishes of an infected individual are risk factors of transmission [33,35,37]. 

Since the smallpox vaccine appears to be at least 85% effective in preventing MPX [3,38], with the JYNNEOS vaccination shown to be effective against MPX, it is now recommended by the US Centers for Disease Control and Prevention (CDC)’s Advisory Committee on Immunization Practices that key personnel, including research laboratory workers and healthcare professionals (HCPs) whose jobs may expose them to Orthopoxviruses, be vaccinated with either ACAM2000 or JYNNEOS [3,31,38,39]. In addition, individuals exposed to the monkeypox virus but not yet received the smallpox vaccine within the last three years are also candidates for vaccination. Based on the CDC’s recommendation, the vaccine should be administered within four days of exposure to prevent the onset of the monkeypox infection; however, if given between 4–14 days after exposure, the vaccine reduces the symptoms of monkeypox [3,31,39]. However, this has to be weighed against potential side effects, with a careful balance of the risks and benefits [31,38]. This is key to the successful administration of the vaccine.

Community pharmacists play crucial roles in vaccine-preventable diseases (VPDs) and the prevention of transmission of infectious diseases during outbreaks. This includes patient education, patient screening, i.e., assessment of the patient’s history and vaccination records, counseling, recommendations, and administration of vaccines [40,41,42]. Community pharmacists are in a unique position to improve access to vaccines and immunization services for the general population, which is crucial during pandemics [43,44,45,46,47]. In addition, current evidence shows that authorizing pharmacists to provide immunization and vaccination services helps to improve vaccination coverage and address the barriers, including addressing vaccine hesitancy and correcting misinformation about the roles of vaccines, their efficacy, and safety [40,45,48,49]. During the COVID-19 pandemic, community pharmacists played a key role in administering COVID-19 vaccines with high patient satisfaction rates alongside generally providing education and advice to the public [44,50,51,52,53,54]. In Saudi Arabia, in 2019, the updated executive regulations of the Practice of Health Professions Law allowed community pharmacists to provide immunization services, including the administration of vaccines [55]. Consequently, during the COVID-19 pandemic, the Saudi Ministry of Health signed a collaboration agreement with a number of community pharmacies to administer COVID-19 vaccines as part of the national COVID-19 vaccination campaign [56]. However, immunization and vaccination services in community pharmacies are still in the infancy stage in a number of countries, including Saudi Arabia. More efforts are needed to address the barriers to implementing and expanding the immunization services and consequently increase public awareness of the services and the benefits to them [44,57,58]. As one of the most accessible healthcare professionals, and based on their active roles in promoting public health in the local communities they are embedded in, community pharmacists with adequate training and up-to-date knowledge can play essential and novel roles in the prevention of emerging infectious diseases, disease outbreaks, and pandemics [41,44,50]. 

Consequently, given the global outbreak of the disease and its community spread in an appreciable number of non-endemic regions, HCPs in non-endemic areas need to be aware of the etiology, epidemiology, transmission modes, and clinical aspects of MPX and other viruses, including potential preventive and treatment measures [3,26,27,31,38,59]. However, to date, we believe only a few studies have been conducted to assess the knowledge of HCPs regarding MPX, its prevention, and management. In addition, those that have been conducted have mainly focused on medical practitioners from a limited number of countries in Asia, Europe, the Middle East, North America, and South America [60,61,62,63,64,65,66,67,68,69], with a few studies including mixed samples of HCPs and healthcare workers (HCWs) [70,71,72,73,74]. This includes Saudi Arabia [60,70,71]. 

Given the currently limited information about community pharmacists’ knowledge of MPX across countries, the aim of this study was to address this gap starting in Saudi Arabia and building on existing studies in Saudi Arabia [60,70,71]. This should provide guidance to key stakeholder groups regarding the development of professional development programs in Saudi Arabia to ensure that qualified pharmacists have evidence-based, up-to-date knowledge of MPX, including its epidemiology, prevention, risk factors, and management, to guide patients. This is because community pharmacists worldwide, including those in Saudi Arabia, are, as mentioned, increasingly playing a key role regarding the prevention and management of infectious diseases and outbreaks, building on their increasing contribution during the recent COVID-19 pandemic [44,50,75,76,77,78,79,80,81,82,83,84]. Key roles of community pharmacists increasingly include addressing misinformation, which has resulted in vaccine hesitancy in a number of countries, as they are often the first HCP that patients consult for a number of conditions, including minor ailments [54,85,86,87,88,89,90,91]. Community pharmacists can also play a vital role in educating travelers to endemic areas regarding current best practices, including preventive measures, to minimize the risk of exposure to themselves. 

## 2. Materials and Methods

### 2.1. Study Design, Setting, and Population

The study is a cross-sectional, questionnaire-based descriptive study. The study population is community pharmacists practicing in the Qassim region, Saudi Arabia. This included Buraidah city and six major governorates in the Qassim region: Unaizah, Al-Rass, Al-Badaya, Riyad AL-Khabra, Al-Methnab, and Al-Bekayriyah. 

The Qassim region was chosen for convenience to the research team. It is located in the central region of Saudi Arabia and had a population of approximately 1.5 million in 2019 [92]. In addition, all major chain pharmacies in Saudi Arabia operate in the Qassim region. Alongside this, the practices, policies, and regulations governing community pharmacy practice in Saudi Arabia are the same in all regions, i.e., national-level policies and regulations. Consequently, we believe this region is representational of Saudi Arabia. The study was conducted from November 2022 to January 2023. 

### 2.2. Sampling Method, Sample Size, and Sample Size Calculations

According to the latest statistics published by the Saudi Ministry of Health, there are a total of 614 community pharmacies in the Qassim region [93]. With a margin of error of 5%, confidence level of 95%, and 50% level of variance, the sample size was estimated to be 237, calculated using the Raosoft^®^ sample size calculator (http://www.raosoft.com/samplesize.html, accesed on 3 October 2022). To compensate for non-responders, the sample size was increased to 250. The selection of the community pharmacies was based on the geographical convenience of the data collectors. In addition, they are located in urban areas of Buraidah city, and the other six governorates included in this study. 

### 2.3. Development of the Questionnaire 

The questionnaire was developed based on an extensive literature search and up-to-date information and facts published by health organizations and authorities across counties. These included the key facts, factsheets for healthcare professionals, and information about monkeypox published by the WHO, US CDC, the European Medicines Agency, the European Centre for Disease Prevention and Control, the Saudi Public Health Authority, and the Saudi Ministry of Health [19,24,28,94,95,96,97]. Previous studies that had explored HCPs and students’ knowledge regarding MPX and its management also informed the development of the questionnaire [64,67,68,98].

The initial version of the questionnaire was given to three experts for their comments, feedback, and suggestions about the face and content validity of the questionnaire. A pilot study was subsequently undertaken with 5 practicing community pharmacists to ensure the suitability, applicability, and clarity of the questionnaire for the target population. Their combined feedback and comments were taken into consideration when finalizing the questionnaire for the principal study. These included suggestions for minor linguistic modifications to ensure clear understanding of all the questions and statements in the questionnaire. The participants also suggested one additional question related to the diagnosis of monkeypox disease to be included. 

The reliability of the questionnaire was subsequently assessed using the dataset of the first 30 participants. The Cronbach’s alpha was 0.753, which is acceptable and indicated a reliable instrument [99,100,101].

The final questionnaire consisted of two parts (Appendix A). The first part consisted of participants’ sociodemographic data, including their gender, age, qualification, and years of experience as a community pharmacist in Saudi Arabia. The second part consisted of 21 statements/questions concerning the current knowledge of community pharmacists regarding monkeypox disease. The 7th question contained 8 items relating to the signs and symptoms of monkeypox. Consequently, the number of questions/items of knowledge was 28. The response options for all statements included ‘yes’, ‘no’, and ‘I do not know’. The correct answer was given 1 point, while an incorrect answer or ‘I do not know’ was given zero points. An ‘I do not know’ answer was given zero points, as it indicates a lack of knowledge. Consequently, the maximum knowledge score was 28. 

The knowledge domain included subdomains/sections incorporating the causative pathogen and epidemiology of MPX (5 statements); diagnosis and clinical characteristics of the disease, i.e., signs, symptoms, and the incubation period; fatality (13 statements); routes of transmission (5 statements); and clinical management and prevention (5 statements). 

### 2.4. Design and Administration of the Questionnaire

The final questionnaire was converted from a paper format into a web-based questionnaire. The web-based survey was designed using an online survey platform (SurveyMonkey, California, 2023—https://www.surveymonkey.com/, accessed on 15 October 2022).

Subsequently, an electronic link and a bar-code were created for access to the web-based questionnaire. Before administration, the web-based survey was tested to ensure there were no technical issues with the access, design, and layout. These included accessing the online survey using different web browsers and by using different devices, i.e., laptops, smartphones, IPads, or tablets, to ensure appropriate layout and design. In addition, as a security feature, only one response was allowed from the same device or IP address [102,103]. 

The research team prepared an invitation letter for the participation of community pharmacists in the study. The letter included a brief overview of the study, its objectives, and the estimated time to answer all the study questions. In the invitation letter, the participants were also provided with instructions on how to access the survey using either a bar-code or a link to directly access the survey. 

The invitation letters were subsequently distributed to community pharmacists by five data collectors. The data collectors were final-year pharmacy students. Before administering the questionnaire, the data collectors were provided with appropriate training on data collection procedures. Following this, they visited the community pharmacies, introduced themselves, and provided the pharmacist with a brief overview about the study as well as the invitation letter, which contained information regarding access to the survey using a bar-code and the link. Finally, they asked the pharmacists to complete the web-based survey at their own convenience. In the invitation letter, the participants were informed that their participation in the study was entirely voluntary and that all the responses would be anonymous and would be reported as aggregated data without identifying any participant. Moreover, as this was an online survey, the participants were informed that their access and submitting their response online was considered as consent to participate in the study.

### 2.5. Data Management and Analysis 

Statistical analyses, including data screening and descriptive and inferential analyses, were performed using IBM SPSS statistics 20.0. Descriptive statistics used to summarize the data included the frequencies, percentages, mean with standard deviations (SD), and median with interquartile range (IQR). Inferential statistics were employed to examine the associations and differences among the study variables. Before the comparison analysis between groups, the normality of the data was checked using tests of normality, i.e., Kolmogorov–Smirnova test, Shapiro–Wilk, Q-Q plots, and box plot [104]. As the data were not normally distributed, non-parametric tests, including Mann–Whitney and Kruskal Wallis tests, were also used. A *p*-value of <0.05 was set as a cut-off point for statistical significance.

### 2.6. Ethics Statement

The study was conducted in accordance with the Declaration of Helsinki. Ethical approval was obtained from the Regional Research Ethics Committee, Qassim region, Saudi Arabia (Approval No. 5643-44-607). 

## 3. Results

### 3.1. Response Rate

A total of 250 community pharmacies were visited by the data collectors. Out of these, nine pharmacists did not agree to participate. In seven community pharmacies, there were two pharmacists present at the time of the visit, while in one pharmacy there were three pharmacists. Consequently, the total number of pharmacists invited was 259. Out of these, 189 electronic responses were received, giving a response rate of 72.97%. All the questionnaires were completed.

### 3.2. Sociodemographic Data

In this study, 86.77% were male participants, with most of the participants ≤ 30 years old (*n* = 97; 51.32%) followed by participants in the age group of 31–40 years (*n* = 69; 36.51%). Twenty-three (12.17%) participants were older than 40 years old. Most of the participants (*n* = 169; 89.42%) had only the entry to the pharmacy profession degree in Saudi Arabia, i.e., BPharm or PharmD degree. In terms of the experience as community pharmacists in Saudi Arabia, 43.39% (*n* = 82) of the participants reported having 1–5 years of experience. The details are summarized in Table 1. 

### 3.3. Knowledge of Monkeypox Disease 

#### 3.3.1. Overall Knowledge of Monkeypox 

The overall knowledge score (mean ± SD) was 17.72 ± 5.56 out of the maximum attainable score of 28 (Table 2), with the overall rate of correct answers for the knowledge statements being 63.29%. Further analysis showed that 6.9% (*n* = 13), 9.5% (*n* = 18), 52.4% (*n* = 99), and 31.2% (*n* = 59) of community pharmacists answered correctly ≤25% (very low knowledge), >25<50% (low knowledge), ≥50–<75% (moderate knowledge), and ≥75% (high knowledge) of the questions, respectively. 

The knowledge subdomain relating to diagnosis and clinical characteristics recorded a high score with a mean ± SD of 8.88 ± 3.0 out of 13 (68.31% correct responses), followed by the knowledge subdomain of routes of transmission of monkeypox virus with a mean ± SD of 3.28 ± 1.48 out of 5 (65.6% correct responses). The total scores of the overall knowledge and each subdomain of knowledge are documented in Table 2 and Figure 1. 

#### 3.3.2. Participants’ Responses to the Knowledge Statements

The vast majority of participants (95.2%) were aware that the causative pathogen of MPX disease was a virus, i.e., the monkeypox virus. However, only 46.6% were aware that MPX is not a newly discovered disease in humans in the 2022 outbreak.

The majority of community pharmacy participants (64.6%) indicated that the current global outbreak is declared by the WHO as a Public Health Emergency of International Concern (PHEIC). In addition, more than half (56.6%) indicated that there were reported cases of MPX in Saudi Arabia in 2022. However, only 29.6% were aware of the current global epidemiological situation in terms of the number of cases reported globally. 

In terms of diagnosis, 64.6% indicated that confirmation of monkeypox infection diagnosis is achieved by using a real-time polymerase chain reaction (PCR) test. In terms of clinical presentations, most were aware of the signs and symptoms, including skin rashes and lesions (87.3%), fever (88.9%), chills (76.7%), lymphadenopathy (74.6%), headache (78.8%), exhaustion/lack of energy (84.7%), muscle aches (83.1%), back pain (65.6%), and respiratory symptoms (64.6%). Moreover, the majority (63%) of participants indicated that MPX symptoms typically last from 2 to 4 weeks. 

However, most participants (77.2%) were not aware of the incubation period of monkeypox disease. Similarly, most participants (66.7%) either incorrectly indicated that the case-fatality rate for monkeypox infection is estimated at approximately 30% or responded with “Ï do not know”. 

In terms of routes of transmission, 85.2% stated that transmission can occur via close contact with skin lesions or respiratory secretions of an infected person with MPX, while only 54.5% stated that it can occur via prolonged face-to-face contact with an infected person. The majority of the community pharmacists (66.7%) stated that people could get infected via direct contact with objects, surfaces, or materials contaminated with monkeypox. In addition, 56.6% stated that the virus could be transmitted from animals to humans via eating inadequately cooked meat and other animal products of an infected animal. In this study, only 12.2% stated that asymptomatic individuals are not the main source of spreading the monkeypox infection. 

In terms of clinical management and prevention, 52.9% of participating community pharmacists stated that medicines such as tecovirimat and brincidofovir that were developed to treat smallpox may be used to treat monkeypox. Similarly, only 42.9% stated that there are approved vaccines for the prevention of monkeypox. According to 70.4% of community pharmacists, standard preventive and precautionary measures are effective in preventing monkeypox, and 67.7% indicated that supportive care and hydration are adequate for patients with an intact immune system. The majority (82.5%) also stated that patients with severe monkeypox symptoms, or those that are at increased risk of developing severe monkeypox infections, should be referred for specialized medical care. The results are summarized in Table 3. 

#### 3.3.3. The Association between Participants’ Knowledge and Their Sociodemographic Characteristics

We examined whether there are differences in the community pharmacists’ overall knowledge in terms of key characteristics, including their gender, age, level of education, and experience. There was only one statistically significant difference, which was their duration of experience as community pharmacists in Saudi Arabia (*p* = 0.040). Community pharmacists with experience between 6 and 10 years recorded significantly higher median knowledge scores compared to others (Table 4). 

## 4. Discussion

We believe this is the first study from Saudi Arabia that has assessed the knowledge of community pharmacists regarding human MPX during the global outbreak in 2022. In fact, we believe this is one of only a few studies in the literature that have specifically assessed the knowledge of pharmacists towards human MPX, its prevention, and treatment, as their role in the healthcare system has further grown following the COVID-19 pandemic [44,79,83,105]. Previous studies have addressed the knowledge and perspectives of medical practitioners across both high-income as well as low- and middle-income countries (LMICs) [60,61,63,65,66,67,68], as well as a mixed sample of HCPs [64,72,73,74], however, not exclusively community pharmacists exploring their knowledge and perspectives.

We were encouraged by a relatively high response rate, i.e., >70%. We believe this is because the data collection procedure using field visits to invite the community pharmacists to participate in the study was very helpful. In addition, community pharmacists were provided with an overview of the study and given the opportunity to submit their responses online at their own convenience, thereby enhancing response rates.

Overall, this study showed that the community pharmacists in the knowledge questions correctly he Qassim region of Saudi Arabia had moderate knowledge of MPX, with the overall rate of correct responses for all knowledge statements being 63.29%. In addition, analysis of the community pharmacists’ knowledge regarding the four aspects/dimensions of the disease showed that they had moderate levels of knowledge, which ranged from 68.31% for diagnosis and clinical characteristics to 58.6% in the domain related to the causative pathogen and epidemiology of MPX. This knowledge level among community pharmacists in Saudi Arabia appears higher than that seen in some previous studies. For example, in Ohio, USA, a study among clinicians documented that on average they answered only 48.9% of the knowledge questions correctly (average knowledge score: 11.23 out of 23) [66]. Similarly, the overall knowledge score among Italian medical practitioners was 51.8% [67]. However, the knowledge level in our study was lower than the findings in Indonesia, where the overall knowledge score among general practitioners was 65.2% (average knowledge score: 13.7 out of 21) [68]. In addition, these were lower than the findings among HCWs in Lebanon, where their overall knowledge score was 69.4% [72]. A high knowledge level was also reported among physicians in Peru, with a median knowledge score of 14 out of 17, and more than 92.9% answered correctly more than 70% of the knowledge items [65]. 

In our study, the vast majority of community pharmacists (95.2%) were aware that MPX is a viral disease, which is similar to the findings among HCPs in other countries, including Saudi Arabia, e.g., 81.1% in Lebanon [72], 94.7% in Saudi Arabia [60], 99.6% in Peru [65], 94.34% in Bangladesh [63], and 97.6% in Indonesia [68]. However, only 46.6% of community pharmacists in our study were aware that MPX is a re-emerging infectious disease and not a newly discovered disease in 2022, which is a concern. Having said this, this was a higher score than that seen among physicians in Turkey, where only 12.7% were aware of this [61]. However, these results were lower than studies in the USA and Italy, in which 91.9% and 95.1% of participants, respectively, were aware that the monkeypox virus is not a newly discovered virus [66,67]. A considerable proportion of community pharmacists in our study were also not fully aware of the epidemiological situation of MPX. This includes the situation in Saudi Arabia and globally, and that MPX is declared by the WHO as a PHEIC. However, this is similar to studies from the USA, Italy, Turkey, Lebanon, and Jordan that reported similar knowledge gaps [61,66,67,72,73]. Consequently, as the community pharmacists are frontline HCPs and easily accessible to the community as reliable sources of information regarding the prevention and management of infectious diseases, it is crucial to provide them with up-to-date knowledge on both emerging and re-emerging infectious diseases. This can help them appropriately educate the public and promote public health strategies to help prevent the spread of any current or future outbreaks of emerging infectious diseases, as well as appropriately manage them. 

Encouragingly, many of the community pharmacists in our study were aware of the confirmation of a monkeypox infection diagnosis, i.e., through PCR tests, and the typical clinical presentations of this disease. This good knowledge could be in part due to the fact that HCPs were typically aware of the role of PCR in the detection of viruses during the COVID-19 pandemic and the common signs and symptoms of viral infections during this unprecedented pandemic. However, most participating community pharmacists were not aware of other specific aspects of MPX, including the incubation period (77.2%) and the case-fatality rate for the infection (66.7%). This is in line with other studies reporting inadequate knowledge regarding these specific aspects [61,74]. Moreover, our findings showed that there is a need to improve community pharmacists’ knowledge in terms of disease management, prevention, vaccination, and modes of transmission. It is crucial that community pharmacists have adequate knowledge about the transmission of the disease via routes other than direct skin contact. In addition, community pharmacists should be aware of the current supportive care and management, including antiviral pharmacotherapy for monkeypox. In addition, awareness of available vaccines against monkeypox for individuals with a high risk of exposure and other preventive measures of transmission of the disease is important. This is because of their increasing role in promoting preventive measures during infectious disease outbreaks, including potentially administering vaccinations. Consequently, tailored educational interventions are important to further update the pharmacists’ knowledge during infectious disease outbreaks, such as the current MPX outbreak and any other outbreaks that could occur in the future. These interventions could be incorporated into current curricula in a timely manner and proactively to ensure the student pharmacists are kept up to date. Innovative virtual education and flexible modules could be utilized to also help with professional development programs and continuing educational activities post qualification. This is particularly important, as a previous national survey showed that the coverage and content of infectious diseases in the pharmacy curricula in Saudi Arabia are not standardized and vary among the pharmacy colleges. For example, the topics of immunization and influenza virus infection are covered as required topics in the curriculum of 83.6 and 80% of the colleges, respectively. However, some topics, including travel medicine and travel-related illnesses, are required topics by only 34.5% of the colleges [106]. This needs to be addressed in light of the recent developments of the pharmacy profession in Saudi Arabia, particularly pharmacists’ authorization to administer vaccines in community pharmacies, and based on the standards of the accreditation commission in Saudi Arabia that requires educational programs, including pharmacy programs, to be periodically reviewed taking into consideration the educational, technical, scientific, and professional developments in the field of practice and specialization [107]. Pharmacy colleges need to ensure new developments, i.e., authorization to administer vaccines and emerging diseases, are covered well and in a timely manner. This includes vaccine-preventable diseases, principles of immunization, new vaccines, and topics related to immunization services at the pharmacies. These include procurement, handling and storage of vaccines, injection and administration techniques, and efficacy and safety of vaccines, as well as any requirements associated with establishing a pharmacy-based immunization program. 

We are aware of a number of limitations with this study. Firstly, the study only included community pharmacists from the Qassim region for the reasons stated earlier. Secondly, it included urban areas in Buraidah city and six major governorates of the region. Rural areas and a few other governorates were not included in the study. In addition, we employed a convenience sampling. However, all these were inevitable due to logistical and practical barriers. Thirdly, the knowledge of HCPs regarding re-emerging infectious diseases in non-endemic countries could be low, especially at the early phases of the outbreaks. This is because these diseases are typically encountered and reported in endemic countries [108]. In addition, these infectious diseases may not be adequately covered in the professional curricula of pharmacy and medical colleges. This is evidenced by the fact that some HCPs had never heard of MPX before the 2022 outbreak [108,109]. Consequently, it is necessary to ensure that such infectious diseases are adequately covered during training, since, as mentioned, pharmacists do play important roles in educating the community as well as travelers about the prevention and transmission of the infections, including vaccinations. Despite these limitations, we believe our study findings are robust in view of our methodology and will provide guidance to health policy makers and academic institutions across countries regarding the training of community pharmacists to enhance their engagement with local populations and travelers regarding the prevention and management of re-emerging infectious diseases, including MPX.

## 5. Conclusions

Overall, community pharmacists in Saudi Arabia had moderate knowledge regarding MPX. However, some gaps in their knowledge regarding this re-emerging disease were identified. Consequently, tailored, flexible, and timely educational interventions are needed to ensure this vital group of HCPs is fully equipped with up-to-date and evidence-based knowledge regarding this viral disease to fully inform patients about its prevention and management. This includes the role of vaccines. We will be monitoring this in the future.

## Figures and Tables

**Figure 1 vaccines-11-00878-f001:**
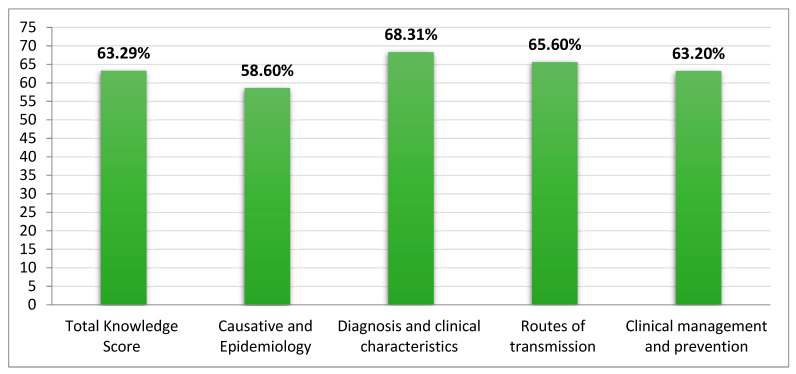
Overall knowledge scores and subdomains. NB: %s refer to mean scores.

**Table 1 vaccines-11-00878-t001:** Sociodemographic data of the participants.

Variable	N (%)
Gender	
Male	164 (86.77)
Female	25 (13.23)
Age group	
24–30	97 (51.32)
31–40	69 (36.51)
41–50	20 (10.58)
>50	3 (1.59)
Educational level	
BPharm/PharmD	169 (89.42)
Postgraduate diploma	8 (4.23)
MSc/MPharm/equivalent or higher	12 (6.35)
Experience as a community pharmacist in Saudi Arabia (in years)	
Less than 1 year	36 (19.05)
1–5	82 (43.39)
6–10	43 (22.75)
More than 10	28 (14.81)

**Table 2 vaccines-11-00878-t002:** The total scores of the overall knowledge and subdomains of knowledge.

Overall Knowledge Scores and Knowledge Subdomains	Mean ± SD
Overall knowledge (28 items)	17.72 ± 5.56 out 28
Causative pathogen and epidemiology of monkeypox (5 items)	2.93 ± 1.20 out of 5
Diagnosis and clinical characteristics (13 items)	8.88 ± 3.0 out of 13
Routes of transmission (5 items)	3.28 ± 1.48 out of 5
Clinical management and prevention (5 items)	3.16 ± 1.48 out of 5

**Table 3 vaccines-11-00878-t003:** Participants’ responses to the knowledge statements *.

No.	Statement	Yes	No	I Do Not Know
K1	Monkeypox is caused by a virus (i.e., a viral disease infection)	**180 (95.2)**	2 (1.1)	7 (3.7)
K2	Monkeypox is a newly discovered disease in humans in 2022	83 (43.9)	**88 (46.6)**	18 (9.5)
K3	Globally, in this year 2022, monkeypox has caused less than 10,000 cases till now	75 (39.7)	**56 (29.6)**	58 (30.7)
K4	In Saudi Arabia, there are reported cases of monkeypox in 2022	**107 (56.6)**	40 (21.2)	42 (22.2)
K5	The current global outbreak of monkeypox is declared by the World Health Organization (WHO) a Public Health Emergency of International Concern	**122 (64.6)**	24 (12.7)	43 (22.8)
K6	Skin rashes and lesions are one of the key characteristics for identifying monkeypox	**165 (87.3)**	12 (6.3)	12 (6.3)
K7	The following are signs and symptoms that can occur in persons with monkeypox:			
	K7.1 Fever	**168 (88.9)**	5 (2.6)	16 (8.5)
	K7.2 Chills	**145 (76.7)**	14 (7.4)	30 (15.9)
	K7.3 Lymphadenopathy (swelling of the lymph nodes)	**141 (74.6)**	16 (8.5)	32 (16.9)
	K7.4 Headache	**149 (78.8)**	19 (10.1)	21 (11.1)
	K7.5 Exhaustion/lack of energy	**160 (84.7)**	10 (5.3)	19 (10.1)
	K7.6 Muscle aches	**157 (83.1)**	8 (4.2)	24 (12.7)
	K7.7 back pain	**124 (65.6)**	25 (13.2)	40 (21.2)
	K7.8 Respiratory symptoms (e.g., sore throat, nasal congestion, cough)	**122 (64.6)**	31 (16.4)	36 (19.0)
K8	The incubation period of monkeypox (i.e., time from exposure to the pathogen to onset of symptoms) can range from 1–5 days	87 (46.0)	**43 (22.8)**	59 (31.2)
K9	Monkeypox symptoms typically last from 2 to 4 weeks	**119 (63.0)**	12 (6.3)	58 (30.7)
K10	The case-fatality rate for monkeypox infection has been estimated at approximately 30%	55 (29.1)	**63 (33.3)**	71 (37.6)
K11	Human-to-human transmission can occur via close contact with skin lesions or respiratory secretions of an infected person with monkeypox	**161 (85.2)**	12 (6.3)	16 (8.5)
K12	Human-to-human transmission can occur via prolonged face-to-face contact with an infected person	**103 (54.5)**	53 (28.0)	33 (17.5)
K13	Persons could get infected via direct contact with objects, surfaces, or materials contaminated with the monkeypox	**126 (66.7)**	36 (19.0)	27 (14.3)
K14	Animal-to-human transmission can occur via eating inadequately cooked meat and other animal products of an infected animal	**107 (56.6)**	43 (22.8)	39 (20.6)
K15	Asymptomatic individuals (i.e., during the incubation period of the infection) are the main source of spreading the monkeypox infection	123 (65.1)	**23 (12.2)**	43 (22.8)
K16	Diagnosis of monkeypox infection is confirmed by using a real-time polymerase chain reaction (PCR) test	**122 (64.6)**	18 (9.5)	49 (25.9)
K17	Medications such as tecovirimat and brincidofovir that were developed to treat smallpox may be used to treat monkeypox	**100 (52.9)**	24 (12.7)	65 (34.4)
K18	There are approved vaccines for the prevention of monkeypox	**81 (42.9)**	51 (27.0)	57 (30.2)
K19	Standard preventive and precautionary measures are effective in preventing monkeypox	**133 (70.4)**	15 (7.9)	41 (21.7)
K20	Supportive care and adequate hydration may be enough for most patients with an intact (healthy) immune system	**128 (67.7)**	24 (12.7)	37 (19.6)
K21	Patients with severe monkeypox symptoms or at increased risk to develop severe monkeypox infection should be referred for a specialized medical care	**156 (82.5)**	9 (4.8)	24 (12.7)

NB: * Correct answers are presented in bold.

**Table 4 vaccines-11-00878-t004:** The association between participants’ knowledge and their sociodemographic characteristics.

Variable	Total Knowledge Score Median (IQR)	*p* Value ^a^
Gender		
Male (*n* = 164)	19 (16–21)	0.500
Female (*n* = 25)	18 (14–21)	
Age group		0.063
24–30 (*n* = 97)	18 (15–21)	
31–40 (*n* = 69)	20 (17–22)	
≥41 (*n* = 23)	19 (16–23)	
Educational level		
BPharm/PharmD (*n* = 169)	19 (16–21)	0.723
Postgraduate diploma (*n* = 8)	16.50 (10–23)	
MSc/MPharm/equivalent or higher (*n* = 12)	20 (12–20)	
Experience as a community pharmacist in Saudi Arabia (in years)		
Less than 1 year (*n* = 36)	18 (14.25–21)	0.040 *
1–5 (*n* = 82)	18 (16–21)	
6–10 (*n* = 43)	20 (18–24)	
More than 10 (*n* = 28)	19 (15–21.75)	

NB: ^a^ Kruskal–Wallis test and Mann–Whitney test were used; * statistically significant at *p* < 0.05.

## Data Availability

Additional data are available on reasonable request from the corresponding author.

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
