# Peer review of "Knowledge of Community Pharmacists in Saudi Arabia Regarding Human Monkeypox, Its Management, Prevention, and Vaccination: Findings and Implications"

_vaccines, 2023, doi:10.3390/vaccines11040878_

Round 1

Reviewer 1 Report

The investigation and research carried out in this work is of great significance, but in view of the survey results, it should be put forward how to improve the current cognition of the disease and prevention measures, and it is suggested to supplement this part.

In addition, the figures and tables in the paper are not standardized, and it is recommended to modify and improveï¼›and the reference format is not standardized enough, it is recommended to modify.

Author Response

Comments and Suggestions for Authors

1) The investigation and research carried out in this work is of great significance, but in view of the survey results, it should be put forward how to improve the current cognition of the disease and prevention measures, and it is suggested to supplement this part.

Author comments: Thank you for your comment. We have now addressed this in the updated version in the Discussion. We hope this is now OK.

2) In addition, the figures and tables in the paper are not standardized, and it is recommended to modify and improveï¼›and the reference format is not standardized enough, it is recommended to modify.

Author comments: Thank you for this. We have re-looked at the format of the Figures and updated this. We will update the Tables and the References with the help of the Journal as we go through the proof stage once the paper has been accepted for publication. We trust this is acceptable.

Reviewer 2 Report

May I kindly ask for more comment about bad answers, low percent od right answers or interpretation of I do not know choice?

Questionnaire is spread among society opinion leaders and advisers on the other hand this group should know how to protect themselve.

Author Response

Comments and Suggestions for Authors

1) May I kindly ask for more comment about bad answers, low percent od right answers or interpretation of I do not know choice?

Author comments: Thank you. We have now addressed this and clarified the interpretation of “I do not know” in the methods section. The response with “I do not know” indicates that the participant is not aware of these facts and lack knowledge - expressed in the statement “I do not know”. I trust this is now OK.

2) Questionnaire is spread among society opinion leaders and advisers on the other hand this group should know how to protect themselves.

Author comments: Thank you for your comment. As seen – this is not always the case across countries and among all HCP groups. We have commented on this further in the updated paper as well as the implications for the future curricula of pharmacists. We hope this is now acceptable.

Reviewer 3 Report

Thank you for the opportunity to review this paper. The authors describe a cross-sectional survey using a convenience sample of community pharmacists in Qassim region, Saudi Arabia. This study is a novel survey of pharmacists with clearly presented results which are of interest to the journal audience. I recommend the manuscript be accepted with minor revisions.

Abstract: Clearly state that this was a convenience sample of community pharmacists in Qassim region, Saudia Arabia.

Introduction: Describe the scope of practice of community pharmacists in Saudi Arabia related to vaccine administration, specifically MPX. Are pharmacists in Qassim region allowed to provide immunization services including administration? Is there separate licensing of pharmacists in Saudi Arabia to administer vaccines? For an international audience with no knowledge of practice in Saudi Arabia, describe the role of pharmacists in Saudi Arabia as it relates to immunization services or lack thereof. 

Methods: Of the 614 community pharmacies in the Qassim region, 250 were visited by the data collectors. Describe the selection of these 250 pharmacies in detail. Was this a geographically convenient sample of the 614 pharmacies? How many were urban versus rural? Were all regions of the Qassim region represented? Were pharmacies visited at different times of day or shifts? 

Results: If pharmacists in Saudi Arabia are permitted to provide immunization services, can you stratify the results based on whether the pharmacists are immunizers?

Discussion:

Are vaccine preventable diseases a required content area in pharmacy school curriculum in Saudi Arabia per accreditation standards? Should they be?

Expand the limitations to address the convenience sample and whether this sample is truly representative of the entire Qassim region.

Author Response

Comments and Suggestions for Authors

1) Thank you for the opportunity to review this paper. The authors describe a cross-sectional survey using a convenience sample of community pharmacists in Qassim region, Saudi Arabia. This study is a novel survey of pharmacists with clearly presented results which are of interest to the journal audience. I recommend the manuscript be accepted with minor revisions.

Author comments: Thank you for these kind words – appreciated!

2) Abstract: Clearly state that this was a convenience sample of community pharmacists in Qassim region, Saudi Arabia.

Author comments: Thank you – now inserted.

3) Introduction: Describe the scope of practice of community pharmacists in Saudi Arabia related to vaccine administration, specifically MPX. Are pharmacists in Qassim region allowed to provide immunization services including administration? Is there separate licensing of pharmacists in Saudi Arabia to administer vaccines? For an international audience with no knowledge of practice in Saudi Arabia, describe the role of pharmacists in Saudi Arabia as it relates to immunization services or lack thereof. 

Author comments: Thank you very much for this comment. We have now addressed this and updated the manuscript accordingly. We hope this is now acceptable.

4) Methods: Of the 614 community pharmacies in the Qassim region, 250 were visited by the data collectors. Describe the selection of these 250 pharmacies in detail. Was this a geographically convenient sample of the 614 pharmacies? How many were urban versus rural? Were all regions of the Qassim region represented? Were pharmacies visited at different times of day or shifts? 

Author comments: We have now addressed this and updated the manuscript accordingly. We hope this is now OK.   

5) Results: If pharmacists in Saudi Arabia are permitted to provide immunization services, can you stratify the results based on whether the pharmacists are immunizers?

Author comments: Pharmacists were recently permitted to provide immunization services in Saudi Arabia (now referenced). However, the immunization services are still in the infancy stage and not fully implemented by community pharmacies in Qassim region due to some current challenges and barriers, e.g. infrastructure for the program, workflow, training of pharmacists, human resources etc.. Consequently, we will be monitoring the situation and address this area of practice in our future studies. We hope this is acceptable.

6) Discussion: Are vaccine preventable diseases a required content area in pharmacy school curriculum in Saudi Arabia per accreditation standards? Should they be?

Author comments: We have now addressed this and updated the manuscript accordingly. We hope this is now OK.   

7) Expand the limitations to address the convenience sample and whether this sample is truly representative of the entire Qassim region.Author comments: Author comments: We have now addressed this and updated the manuscript accordingly. We hope this is now acceptable.
